

# Development of a noninvasive photograph-based method for the evaluation of body condition in free-ranging brown bears

Yuri Shirane[1], Fumihiko Mori[1], Masami Yamanaka[2], Masanao Nakanishi[2], Tsuyoshi Ishinazaka[2], Tsutomu Mano[3], Mina Jimbo[1], Mariko Sashika[1], Toshio Tsubota[1] and Michito Shimozuru[1]

[1] Graduate School of Veterinary Medicine, Hokkaido University, Sapporo, Hokkaido, Japan
[2] Shiretoko Nature Foundation, Shari, Hokkaido, Japan
[3] Hokkaido Research Organization, Sapporo, Hokkaido, Japan

Corresponding author
Michito Shimozuru,
shimozuru@vetmed.hokudai.ac.jp

## ABSTRACT

Body condition is an important determinant of health, and its evaluation has practical applications for the conservation and management of mammals. We developed a noninvasive method that uses photographs to assess the body condition of free-ranging brown bears (*Ursus arctos*) in the Shiretoko Peninsula, Hokkaido, Japan. First, we weighed and measured 476 bears captured during 1998–2017 and calculated their body condition index (BCI) based on residuals from the regression of body mass against body length. BCI showed seasonal changes and was lower in spring and summer than in autumn. The torso height:body length ratio was strongly correlated with BCI, which suggests that it can be used as an indicator of body condition. Second, we examined the precision of photograph-based measurements using an identifiable bear in the Rusha area, a special wildlife protection area on the peninsula. A total of 220 lateral photographs of this bear were taken September 24–26, 2017, and classified according to bear posture. The torso height:body/torso length ratio was calculated with four measurement methods and compared among bear postures in the photographs. The results showed torso height:horizontal torso length (TH:HTL) to be the indicator that could be applied to photographs of the most diverse postures, and its coefficient of variation for measurements was <5%. In addition, when analyzing photographs of this bear taken from June to October during 2016–2018, TH:HTL was significantly higher in autumn than in spring/summer, which indicates that this ratio reflects seasonal changes in body condition in wild bears. Third, we calculated BCI from actual measurements of seven females captured in the Rusha area and TH:HTL from photographs of the same individuals. We found a significant positive relationship between TH:HTL and BCI, which suggests that the body condition of brown bears can be estimated with high accuracy based on photographs. Our simple and accurate method is useful for monitoring bear body condition repeatedly over the years and contributes to further investigation of the relationships among body condition, food habits, and reproductive success.

## INTRODUCTION

Body condition, defined as the energetic state in an individual, especially the relative size of energy reserves such as fat and protein (*Gosler, 1996*; *Schulte-Hostedde, Millar & Hickling, 2001*; *Peig & Green, 2009*), is an important determinant of health in both terrestrial and marine mammals. It serves as an indicator of food quality (*Mahoney, Virgl & Mawhinney, 2001*; *McLellan, 2011*), reproductive success (*Noyce & Garshelis, 1994*; *Guinet et al., 1998*), and survivorship (*Young, 1976*; *Gaillard et al., 2000*). Animals in good body condition generally have more energy reserves and are therefore more resilient and more likely to survive than those in poorer condition (*Cook et al., 2004*; *Clutton-Brock & Sheldon, 2010*). In females, reproductive traits such as litter mass, number of litters, neonatal mass, and breeding life-span increase with body condition (*Samson & Huot, 1995*; *Atkinson & Ramsay, 1995*). Therefore, evaluating body condition is of general biological interest but also has practical applications for the conservation and management of mammals.

The body condition of living mammals has been assessed with morphometric measurements (*Guinet et al., 1998*; *Cattet et al., 2002*), blood analyses (*Hellgren, Rogers & Seal, 1993*; *Gau & Case, 1999*), bioelectrical impedance (*Farley & Robbins, 1994*; *Hilderbrand, Farley & Robbins, 1998*), and ultrasound measurements of subcutaneous fat (*Morfeld et al., 2014*). However, these methods are unsuitable as a routine method because they require repeated capture of individuals. Applying these methods to free-ranging, large-bodied mammals is inherently difficult because the capture operation is dangerous for researchers and may affect animal behavior and survival through anesthesia and direct handling. An alternative, noninvasive evaluation method is body condition scoring (BCS). BCS is a subjective assessment of subcutaneous body fat stores based on a visual or tactile evaluation of muscle tone and key skeletal elements (*Otto et al., 1991*; *Burkholder, 2000*). Various BCS systems have been established for monitoring individual condition in companion animals (e.g., dogs and cats: *Laflamme, 2012*), livestock (e.g., cattle, horses, and pigs: *Wildman et al., 1982*; *Henneke et al., 1983*; *Department for Environment Food and Rural Affairs, 2004*), and also wildlife (e.g., bears, dolphins, and elephants: *Stirling, Thiemann & Richardson, 2008*; *Morfeld et al., 2014*; *Joblon et al., 2015*). In addition, visual assessment criteria based on photographs have been used to evaluate relative body condition in whales. Photograph-based measurements of the length and width of gray whales (*Eschrichtius robustus*) from vertical aerial photogrammetry can reveal changes in body condition associated with fasting during winter migrations (*Perryman & Lynn, 2002*). These studies demonstrate that it is possible to visually detect changes in body condition without capturing animals.

For killed or captured bears (*Ursus* spp.), a body condition index (BCI) has been established based on residuals from the regression of body mass against straight-line body length (i.e., the observed mass minus the expected mass: *Cattet et al., 2002*). Independently of sex or age, the BCI has a strong positive relationship with true body condition, defined as the combined mass of fat and skeletal muscle relative to body size (*Atkinson, Nelson & Ramsay, 1996*; *Cattet et al., 2002*). The BCI has higher positive values for bears in better condition and lower negative values for those in poorer condition. In addition, predictive

equations have been developed to estimate body mass and condition in bears from measurements of straight-line body length and axillary girth (*Bartareau, 2017*; *Moriwaki et al., 2018*). However, to clarify seasonal and annual changes in the body condition of bears, it is necessary to develop a method that can be used to monitor body condition repeatedly and continued for several years. For proper conservation and management of bear populations, it is important to develop a noninvasive method of assessing body condition in bears without capture operations.

In this study, we developed a noninvasive method of evaluating the body condition of brown bears (*Ursus arctos*) based on morphometric measurements obtained from photographs. Brown bears are large omnivores that can change their diet in response to spatial and seasonal variation in food resources (*Bojarska & Selva, 2012*) and have a wide distribution throughout the Northern Hemisphere. In Japan, they occur only on Hokkaido, the northernmost island of the country (Fig. 1). Our goal was to develop an accurate, photograph-based evaluation method that could be applied to bears in various postures. To achieve this, we took the following three steps. First, we conducted preliminary analyses using BCIs calculated from actual measurements of killed or captured bears to obtain fundamental information on the body condition of Hokkaido brown bears. We also investigated whether the ratio of torso height to body length could be used as an indicator of body condition by examining its correlation with BCI. Second, we validated the precision of photograph-based measurements using photographs of an identifiable female. We identified four candidate methods of measurement, including horizontal body length, Euclidean body length, polygonal-line body length, and horizontal torso length. Then, we examined which method had the largest number of applicable photographs with sufficiently small variation in measurement. We also examined the ability of our method to detect seasonal changes in body condition. Third, we validated the accuracy of the photograph-based measurement method by examining the correlation between BCIs calculated from actual measurements of captured individuals and photographic evaluation of the same individuals.

## MATERIALS & METHODS

### Study area

This study was conducted in the Shiretoko Peninsula (43°50′–44°20′N, 144°45′––145°20′E), Hokkaido, Japan (Fig. 1). This peninsula has one of the largest brown bear populations worldwide (*Hokkaido Government, 2017*), and an area from the middle to the tip of the peninsula has been designated as Shiretoko National Park and a UNESCO World Natural Heritage Site. During 1998–2017, we collected body masses and morphometrics from brown bears captured for research purposes, killed for nuisance control, or harvested from the peninsula, including the towns of Shari and Rausu (Fig. 1). In addition, a focal survey was conducted in the Rusha area (44°12′N, 145°12′E; Fig. 1), a special wildlife protection area. Public access is not allowed without permission and there is no human residence except for one fishermen's settlement. Because the fishermen have not excluded bears from the settlement area in the last few decades, the bears have become habituated

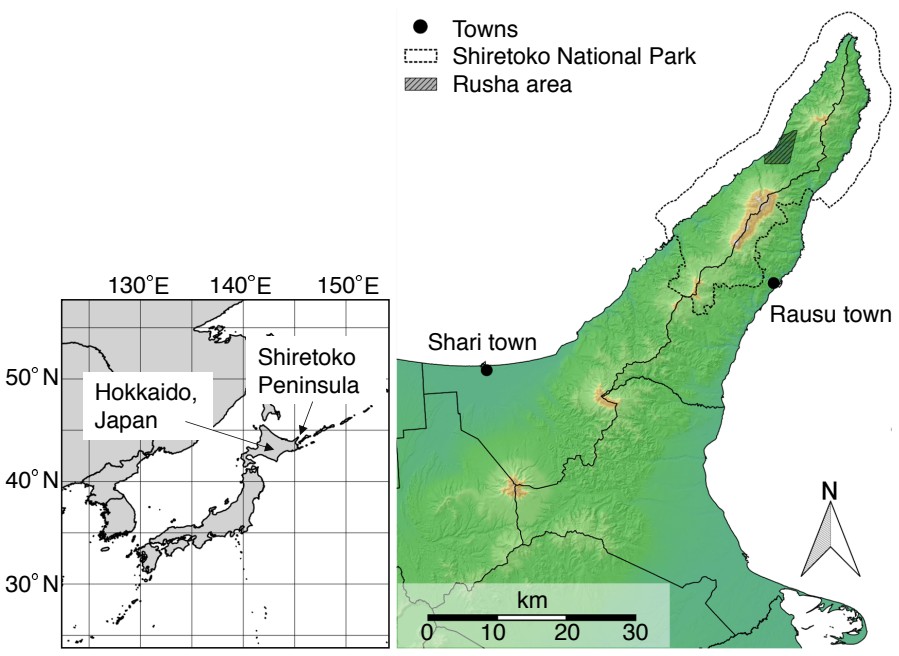

**Figure 1** **Map of the Shiretoko Peninsula, Hokkaido, Japan.** The dotted line indicates the Shiretoko National Park. This map was created using QGIS version 2.16 (http://qgis.osgeo.org) and edited by the author. The base-map image, contour lines, topographic features are based on the National Land Numerical Information published by National Spatial Planning and Regional Policy Bureau, Ministry of Land, Infrastructure, Transport, and Tourism of Japan (available from http://nlftp.mlit.go.jp/ksj/index.html, accessed 7 December 2017).

to the existence of humans, which enables direct observation at close range. Long-term monitoring of identifiable bears has been conducted in this region since 2006 (*Shimozuru et al., 2017*). We collected body masses, morphometrics, and photographs of female bears in the Rusha area during 2014–2016. Field experiments were approved by Hokkaido Regional Environment Office and Kushiro Nature Conservation Office (Permit Number: 1606091 and 1705182).

## Bear capture and measurements

Bears were sampled each year during 1998–2017 between April and November. Most samples were obtained from bears killed for nuisance control or harvested, and some were obtained from bears captured for research purposes. The variables recorded for each bear included an identification code, date of measurement, location, body mass (kg), and straight-line body length (cm) (Data S1). Body mass was measured with calibrated hanging spring scales. Body length was measured with a non-stretchable tape measure as the straight-line distance from the tip of the nose to the end of the last tail vertebra while the bear was aligned laterally. In addition, we measured torso height (cm) as the distance from the lowest point of the abdomen to the spine in females ≥5 years old during 2014–2017. We also collected tissue (e.g., muscle and liver) from killed bears and blood and hair samples from captured bears for DNA extraction, which allowed us to identify

individuals and their sex (*Shimozuru et al., 2017*; *Shirane et al., 2018*). Among 503 killed or captured individuals, 22 individuals were sampled more than once during our study period due to repeated capture or killing after capture; we used only the measurement taken at the greatest age in the following analyses.

We estimated the age in years of most bears captured or killed by counting the cementum annuli of the teeth (*Yoneda, 1976*). For some individuals, we could not determine the exact age due to many cementum-layers developed in old individuals or poor quality of teeth samples. Individuals whose age range could only be estimated were excluded from the growth curve analyses but were included for BCI and subsequent analyses if the growth curve results (detailed below) allowed their classification into an age class. For example, females ≥5 years old were excluded from growth curve analyses but were used as adults for subsequent analyses, whereas males ≥5 years old were excluded from all analyses.

All bears were captured live in accordance with the Guidelines for Animal Care and Use of Hokkaido University and all procedures were approved by the Animal Care and Use Committee of the Graduate School of Veterinary Medicine, Hokkaido University (Permit Number: 1152 and 15009). The protocols for capture received annual approval from the Ministry of the Environment, Japan, and the Hokkaido Government through research permit applications.

## Growth curve of body length

To estimate the age at which the growth of body length was completed, growth pattern in body length was examined using a von Bertalanffy curve as previously described in bears (*Kingsley, Nagy & Reynolds, 1988*; *Derocher & Stirling, 1998*; *Derocher & Wiig, 2002*; *Bartareau, Cluff & Larter, 2011*). The von Bertalanffy size-at-age equation was used in the form $A_t = A_\infty(1 - e^{-K(t-T)})$, where $A_t$ is body length (in cm) at age t, $A_\infty$ is asymptotic body length (in cm), K is a size growth rate constant (year$^{-1}$ ), and T is a fitting constant (extrapolated age at zero size; in years). We conducted F tests to determine whether the parameters of the von Bertalanffy growth equation differed significantly by sex. We conducted analyses using FSA package version 0.8.30 (*Ogle, Wheeler & Dinno, 2020*) and nlstools package version 1.0-2 (*Baty et al., 2015*) in R (*R Core Team, 2019*). According to the age reaching 95% of the asymptotic body lengths obtained from this analysis (detailed below), bears were assigned to three age classes for each sex: cubs (0–1 year old), subadults (age 1–4 years and 1–7 years for females and males, respectively), and adults (age ≥5 years and ≥8 years for females and males, respectively).

## BCI of killed or captured bears

We calculated BCI as previously described in *Cattet et al. (2002)*. Specifically, body mass and length values were transformed to natural logarithms and a least-squares linear regression analysis was conducted to describe the relationship between the ln-transformed values. The standardized residuals of this linear regression were used as BCI. In addition, as a preliminary experiment for the evaluation of body condition using photographs, we calculated the ratio of torso height to body length (TH:BL) using actual measurement data.

*Statistical methods.*—To determine if the BCI was independent of body size, we investigated the correlation between BCI and body length that is an indicator of body
size (*Mahoney, Virgl & Mawhinney, 2001*; *Cattet et al., 2002*). BCI was compared among seasons and age-sex classes using two-way analysis of variance (ANOVA). We used Tukey multiple comparisons (*Tukey, 1977*) to evaluate differences between the mean values of each comparison. Based on major changes in diet (*Ohdachi & Aoi, 1987*), we divided the sampling period into three seasons: spring (April to June; main diet of grass), summer (July and August; main diet of grass and ants), and autumn (September to November; main diet of berries and acorns). In addition, we linearly regressed BCI on the TH:BL of the same individuals and calculated the correlation coefficient. We also used correlation analysis between TH:BL and body length to investigate the effects of body size. We conducted all statistical analyses in R (*R Core Team, 2019*).

## Obtaining and filtering of photographs

Periodic surveys ($\geq 1$ day/2 weeks) have been conducted since 2011 in the Rusha area, mainly for monitoring the reproductive status of identifiable females (*Shimozuru et al., 2017*). This area is a narrow estuarine coast stretching south to north for approximately 3 km. Field teams patrolled the area by car and waited for bears to emerge from the vegetation on the mountainside. When bears appeared, we followed individuals, maintaining a distance of about 20–100 m. Individual bears were identified by field staff according to their appearance as described in *Shimozuru et al. (2017)*, and close-up photographs were taken from multiple angles with a digital, single-lens reflex camera (Nikon D800, NIKON Co., Tokyo, Japan; or Canon EOS 5D, Canon Inc., Tokyo, Japan).

For each survey in the Rusha area, lateral photographs of each individual bear were selected and graded based on several attributes: camera focus, camera tilt (vertical), camera angle (horizontal), body/torso height measurability, and body/torso length measurability for photography; and degree of body arch (vertical), straightness of body (horizontal), degree of neck flexing (vertical), and degree of neck bending (horizontal) for bear posture (Table S1, Fig. S1). Each photograph was given a score of 1 (good quality), 2 (medium quality), or 3 (poor quality) for each attribute. Photographs that were given a score of 3 for any attribute were removed from further analyses.

## Morphometric measurements from photographs

We used ImageJ version 1.52a (*Schneider, Rasband & Eliceiri, 2012*) to extract morphometric measurements from lateral photographs of bears. We first adjusted the angle of the photographs according to the ground surface, then measured the torso height in pixels (TH) as the distance perpendicular to the ground from the lowest point of the abdomen to the highest point of the waist (Fig. 2). Length measurements (in pixels) included the following four methods: the horizontal straight-line body length (HBL, Fig. 2) was the straight-line distance from the tip of the nose to the base of the tail; the Euclidean straight-line body length (EBL, Fig. 2) was the Euclidean distance from the base of the tail to tip of the nose; the polygonal-line body length (PBL, Fig. 2) was the sum of the distance from the base of the tail to the highest part of the shoulder parallel to the ground surface, from that point to the base of the ear, and from that point to the tip of the nose; and the horizontal straight-line torso length (HTL, Fig. 2) was the straight-line distance

**Figure 2** Four candidate methods of measurement to evaluate the body condition of brown bears in the Shiretoko Peninsula, Hokkaido, Japan. (A) Horizontal body length (HBL). (B) Euclidean body length (EBL). (C) Polygonal-line body length (PBL). (D) Horizontal torso length (HTL). Photo credit: Yuri Shirane.

from the base of the tail to the highest part of the shoulder parallel to the ground. For all measurements, any area that could be clearly judged to be only fur was excluded from the measurement range.

## Precision of measurements from photographs

To examine the precision of each photograph-based measurement method and the effects of bear posture, we used photographs of one bear (bear ID: HC) that was monitored routinely in the Rusha area during 2016–2018. We classified photographs according to bear posture (Table S1 and Fig. S1): photographs that had a score of 1 for all attributes were assigned to "Good", those with a score of 2 for body straightness only were assigned to "BS", those with a score of 2 for neck flexing only were assigned to "NF", and those with a score of 2 for neck lateral bending only were assigned to "NB". Photographs that were not assigned to any category were excluded from these analyses.

First, to determine the number of measurements sufficient to reduce measurement error, we assessed measurement precision within photographs by repeatedly measuring (50 times) the body morphometrics from the best photograph taken on September 25, 2017, and assigned to the "Good" category. From these measurements, the coefficients of variation (CVs) for TH, HBL, EBL, PBL, HTL, and the ratio of TH to body/torso length were calculated. In addition, by considering the standard deviation obtained from the 50 measurements as the population standard deviation, we calculated the measurement error at a given number of measurements. We ultimately adopted the minimum number of measurements for which the measurement error had a value that did not affect the

second decimal place (i.e., <0.0025). In the following analyses, TH and body/torso length were measured three times, and the TH:body/torso length ratio was calculated from the respective average values according to our results (detailed below).

Second, we assessed measurement precision between bear postures (differences between repeated measures of the same individual taken from photographs with different postures) by taking measurements from photographs in different posture categories. To eliminate the effects of seasonal changes in body condition, we restricted these analyses to photographs taken September 24–26, 2017. The TH:body/torso length ratio was compared among the posture categories for each measurement method with one-way ANOVA. We used Tukey multiple comparisons (*Tukey, 1977*) to evaluate differences between the mean values of different categories. Then we calculated the CV of each method using all of the photographs applicable to the method to evaluate the measurement precision of each method. We compared CVs among the four methods using an asymptotic test (*Feltz & Miller, 1996*). From these results, we adopted the method that could be applied to photographs of the most diverse postures while maintaining a sufficiently high measurement precision between photographs (CV <5%). In accordance with these results (detailed below), we used TH:HTL as an indicator of body condition in the following analyses.

Third, to examine whether TH:HTL reflected seasonal changes in body condition, we used photographs taken between late June and early October during 2016–2018. For each half-month, the best two or more photographs were selected and the median TH:HTL obtained from these photographs was considered the evaluation value for that half-month. We compared TH:HTL among half-months using one-way ANOVA and used Tukey multiple comparisons (*Tukey, 1977*) to evaluate differences between the mean values of each half-month. We conducted statistical analyses using Microsoft Excel® (*Microsoft Corporation, 2016*) or R (*R Core Team, 2019*).

### Accuracy of measurements from photographs

We examined the accuracy of photograph-based measurement methods using actual measurement data for seven females (≥5 years old) captured in the Rusha area (bear IDs: BE, DR, GI, KR, LI, RI, and WK). We collected photographs of these individuals from within 3 days before and after the days the individuals were captured. After filtering the photographs, we measured TH and HTL and calculated the TH:HTL ratio using two or more of the best photographs. We also calculated BCI using the body mass and length measured at the time of capture.

*Statistical methods.*—We linearly regressed BCI on the TH:HTL ratio and calculated the correlation coefficient. We conducted statistical analyses using Microsoft Excel® (*Microsoft Corporation, 2016*).

## RESULTS

We weighed and measured 503 different individuals: 9 females from the Rusha area during 2014–2016 and 494 individuals (201 females and 293 males) from other parts of the Shiretoko Peninsula during 1998–2017. Among these, we assigned an age (in years) to 432 individuals (174 females and 258 males) and an age range to 56 individuals.

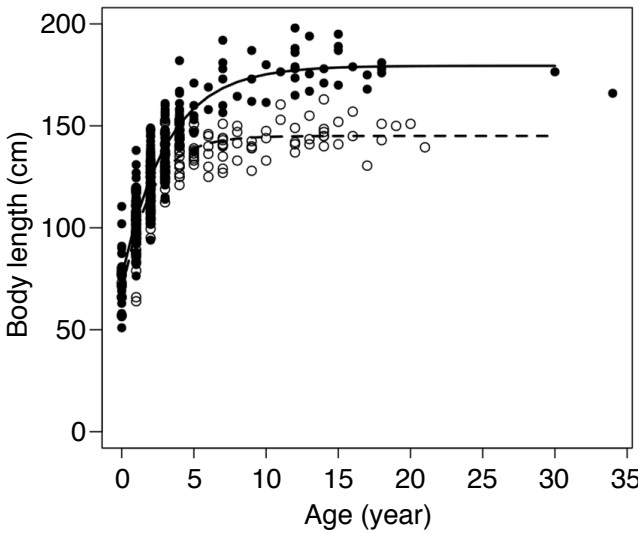

**Figure 3** **Body length at age for 174 female (○) and 258 male (●) brown bears in the Shiretoko Peninsula, Hokkaido, Japan.** Fitted lines represent the von Bertalanffy growth curve for females (dashed line) and males (solid line).

**Table 1** **Parameter estimates (±SE) for von Bertalanffy size-at-age curves for the body lengths of 432 brown bears in the Shiretoko Peninsula, Hokkaido, Japan.** $A_{\infty}$ is the asymptotic body length, K is the size growth constant, and T is the theoretical age at which the animal would have size 0.

| Sex | $A_{\infty}$ (cm) | K (year$^{-1}$) | T (years) | $n$ |
|---|---|---|---|---|
| Female | 145.07 ± 1.48 | 0.51 ± 0.04 | −1.28 ± 0.16 | 174 |
| Male | 179.47 ± 2.39 | 0.32 ± 0.02 | −1.73 ± 0.14 | 257 |

## Body length growth curves

von Bertalanffy curves were successfully fitted to body length data for the 432 individuals with age (in years) assignments (Fig. 3, Table 1, Data S1). The growth curves differed significantly by sex ($F_{3,426} = 76.63$, $p < 0.001$). Females had achieved 95% of their asymptotic body length at 4.6 years of age, whereas males took 7.6 years to reach the same proportion. In accordance with these results, 476 individuals, including those with known age ranges, were classified into age classes and used in the subsequent analyses: 8 females and 19 males were cubs, 105 females (1–4 years) and 211 males (1–7 years) were subadults, and 92 females ≥5 years old and 41 males ≥8 years old were adults.

## BCI of killed or captured bears

Natural logarithmic transformation of the body mass and length data resulted in a linear relationship between mass and length as follows: ln body mass = 3.04 ● ln body length − 10.41 ($R^2 = 0.94$, residual standard deviation = 0.19, Fig. 4, Data S1). To facilitate estimation of BCI for brown bears, we developed the following model: BCI = (ln body mass − 3.04 ln body length + 10.41)/0.19. There was no correlation between body length
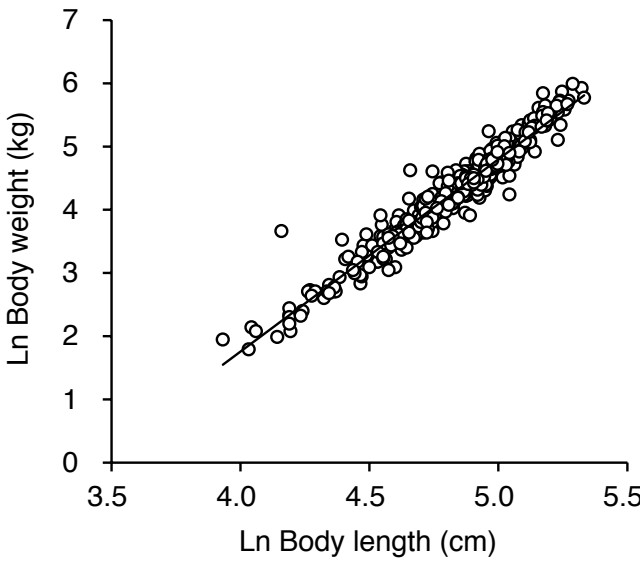

**Figure 4** **Relationship between ln-transformed body weight and ln-transformed body length for 476 brown bears killed or captured in the Shiretoko Peninsula, Hokkaido, Japan, during 1998–2017.** The solid line indicates the best fitting line determined by ordinary least squares regression and is described as follows: ln body weight $= 3.04 \bullet$ ln body length $-10.41$ ($R^2 = 0.94$, residual standard deviation $= 0.19$).

and BCI ($r = 0.037$, $p = 0.39$), which indicates that BCI was independent of body size (Fig. S2).

An ANOVA of BCI showed that BCI varied significantly by season ($F_{2,459} = 13.26$, $p < 0.001$; Table 2, Fig. 5), with bears sampled in spring and summer having lower BCI than bears sampled in autumn (both $p < 0.001$). Differences among age-sex classes were also significant ($F_{5,459} = 4.20$, $p < 0.001$): Adult males showed higher BCI than adult females ($p = 0.002$), subadult females ($p < 0.001$), and subadult males ($p = 0.003$), whereas BCI did not differ among other age-sex classes ($p = 0.35 - 0.99$). The interaction between season and age-sex class was not significant ($F_{9,459} = 0.46$, $p = 0.90$).

We obtained measurements of torso height from 23 adult females. A positive correlation was found between the TH:BL ratio and BCI ($r = 0.81$, $p < 0.001$; Fig. 6, Data S1). There was no correlation between body length and TH:BL ($r = -0.068$, $p = 0.73$), which indicates that TH:BL was independent of body size (Fig. S3).

## Precision of measurements from photographs

A total of 220 photographs of the same bear (bear ID: HC) were taken September 24–26, 2017. After filtering based on photographic conditions and the body arch of the bear (Table S1 and Fig. S1), 101 photographs remained. Of these photographs, 15 were assigned to "Good", 9 to "BS", 10 to "NF", and 9 to "NB".

Based on 50 repeat measurements of the best photograph in the "Good" category, the CV in measurement error within photographs was estimated to be 0.29% for torso height and 0.27%, 0.29%, 0.26%, and 0.45% for HBL, EBL, PBL, and HTL, respectively. For all

**Table 2 Mean body condition index (BCI) and body weight of brown bears in six age-sex classes captured and measured in the Shiretoko Peninsula, Hokkaido, Japan, during 1998–2017.** Spring is April–June, summer is July and August, and autumn is September–November.

| | Spring | | | Summer | | | Autumn | | |
|---|---|---|---|---|---|---|---|---|---|
| **Class** | **BCI** | **Weight (kg)** | ***n*** | **BCI** | **Weight (kg)** | ***n*** | **BCI** | **Weight (kg)** | ***n*** |
| Female | | | | | | | | | |
| Adult | −0.39 ± 0.26 | 98.5 ± 4.8 | 14 | −0.18 ± 0.12 | 101.4 ± 3.6 | 35 | 0.20 ± 0.14 | 116.2 ± 4.1 | 43 |
| Subadult | −0.39 ± 0.13 | 61.9 ± 5.4 | 27 | −0.20 ± 0.21 | 53.3 ± 4.1 | 46 | 0.17 ± 0.16 | 72.5 ± 6.2 | 32 |
| Cub | – | – | 0 | 0.36 ± 0.48 | 10.8 ± 1.7 | 3 | −0.08 ± 0.25 | 16.1 ± 0.7 | 5 |
| Male | | | | | | | | | |
| Adult | 0.63 ± 0.35 | 230.1 ± 13.1 | 4 | 0.37 ± 0.14 | 213.4 ± 7.2 | 24 | 1.16 ± 0.19 | 309.2 ± 13.2 | 13 |
| Subadult | −0.15 ± 0.12 | 78.8 ± 4.0 | 77 | −0.01 ± 0.08 | 85.2 ± 5.5 | 91 | 0.51 ± 0.14 | 99.7 ± 6.6 | 43 |
| Cub | −0.30 ± 0.00 | 6.0 ± 0.0 | 1 | 0.12 ± 0.45 | 11.5 ± 1.6 | 6 | 0.16 ± 0.33 | 22.4 ± 4.3 | 12 |
| All classes pooled | −0.20 ± 0.00 | 81.7 ± 0.1 | 123 | −0.03 ± 0.00 | 92.5 ± 0.1 | 205 | 0.36 ± 0.00 | 107.9 ± 0.1 | 148 |

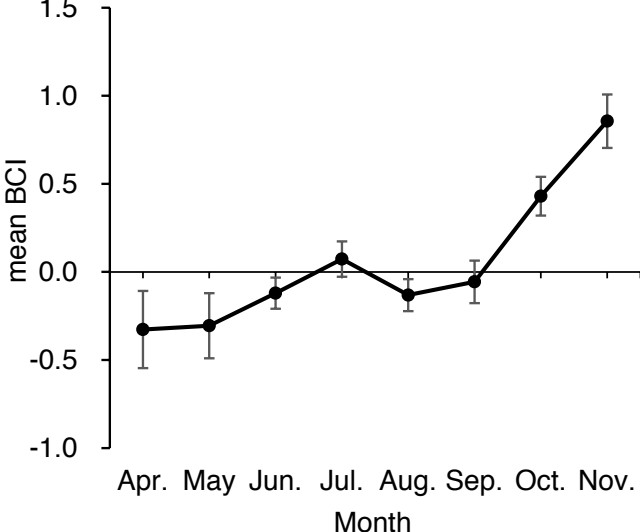

**Figure 5 Monthly mean body condition index (BCI) of 476 brown bears killed or captured in the Shiretoko Peninsula, Hokkaido, Japan, during 1998–2017.** Error bars show SEs.

measurement methods, we reduced the measurement error of the ratio of height to length to less than ±0.0025 by measuring height and body/torso length ≥3 times (Table 3).

The torso height:body/torso length ratio differed among the posture categories for all measurement methods ($p < 0.001$ for TH:HBL and TH:EBL, $p = 0.005$ for TH:PBL, and $p = 0.002$ for TH:HTL, Table 4, Data S2). TH:HBL and TH:EBL obtained from photographs in the "BS", "NF", and "NB" categories differed significantly from the results obtained from photographs in the "Good" category (Table 4). TH:PBL measured using "BS" and "NB" photographs were different from those of "Good" photographs (Table 4). TH:HTL differed from "Good" photographs only when we used "BS" photographs (Table 4). When we used all photographs in each category that did not differ from "Good" for each method,

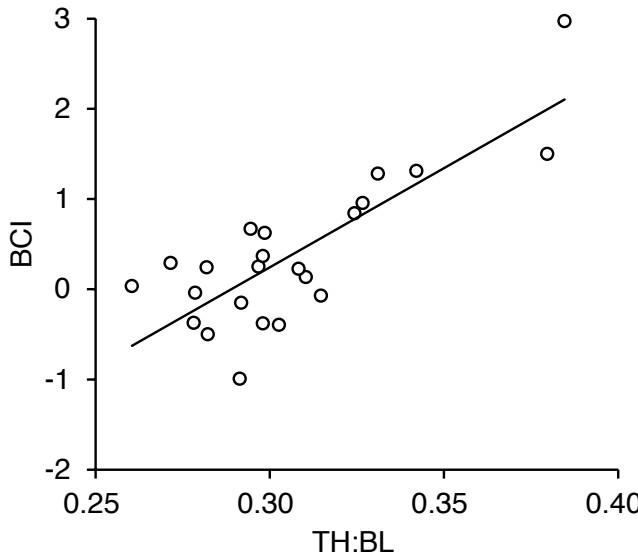

**Figure 6** Relationship between torso height:body length ratio (TH:BL) and body condition index (BCI) for 23 adult female brown bears killed or captured in the Shiretoko Peninsula, Hokkaido, during 2014–2017. Pearson's correlation was $r = 0.81$ ($R^2 = 0.65$, $p < 0.001$).

**Table 3** Measurement precision within photographs of an adult female brown bear (bear-ID: HC) in the Rusha area of the Shiretoko Peninsula, Hokkaido, Japan. The standard error (SE) in the ratio of torso height to body/torso length at a certain number of measurements was calculated by considering the standard deviation (SD) obtained from 50 times measurements as the population standard deviation. CV means coefficient of variation.

| Methods | 50 measurements | | SE (number of measurement) | | |
| --- | --- | --- | --- | --- | --- |
| | Mean ± SD | CV | (Two) | (Three) | (Four) |
| TH:HBL | 0.4316 ± 0.0016 | 0.36% | 0.0011 | 0.0009 | 0.0008 |
| TH:EBL | 0.4266 ± 0.0015 | 0.35% | 0.0010 | 0.0009 | 0.0007 |
| TH:PBL | 0.4163 ± 0.0015 | 0.35% | 0.0010 | 0.0009 | 0.0007 |
| TH:HTL | 0.7504 ± 0.0040 | 0.53% | 0.0028 | 0.0023 | 0.0020 |

Notes.
TH, torso height; HBL, horizontal body length; EBL, Euclidean body length; PBL, polygonal line body length; HTL, horizontal torso length.

the CV was <5% for all methods and did not differ among methods ($p = 0.067$): 2.47% in TH:HBL (photo $n = 15$), 2.19% in TH:EBL ($n = 15$), 3.18% in TH:PBL ($n = 25$), and 3.93% in TH:HTL ($n = 34$). Given these results, TH:HTL was adopted as the measurement method with both the largest number of applicable photographs and a CV <5% (i.e., high measurement precision between photographs).

By calculating TH:HTL using photographs of the same bear (bear ID: HC) taken from late June to early October during 2016–2018, we determined that TH:HTL reached its lowest in late August ($0.567 \pm 0.012$; mean ± SE) and its highest in early October ($0.714 \pm 0.015$, Fig. 7). TH:HTL varied significantly among half-months ($F_{7,16} = 18.41$, $p < 0.001$) and was lower in late August than in late June ($p = 0.013$), early July ($p = 0.007$), late July

Shirane et al. (2020), *PeerJ*, DOI 10.7717/peerj.9982

**Table 4  Mean (±SD) ratio of torso height to body/torso length obtained from photographs of an adult female brown bear (bear ID: HC) in the Rusha area of the Shiretoko Peninsula, Hokkaido, Japan.** *P* values are based on comparisons of mean ratios from the "Good" category versus other categories for each measurement method with Tukey multiple comparisons. Bold characters indicate significant differences. The "Good" category contained photographs with a score of 1 for all attributes, "BS" had a score of 2 for body straightness only, "NF" had a score of 2 for neck flexing only, and "NB" had a score of 2 for neck lateral bending only.

| Categories | *n* | TH:HBL | | | TH:EBL | | | TH:PBL | | | TH:HTL | | |
|---|---|---|---|---|---|---|---|---|---|---|---|---|---|
| | | mean ± SD | CV | *p*-value | mean ± SD | CV | *p*-value | mean ± SD | CV | *p*-value | mean ± SD | CV | *p*-value |
| Good | 15 | 0.416 ± 0.010 | 2.47% | | 0.409 ± 0.009 | 2.19% | | 0.394 ± 0.011 | 2.88% | | 0.711 ± 0.025 | 3.45% | |
| BS | 9 | 0.454 ± 0.020 | 4.34% | **<0.001** | 0.431 ± 0.017 | 4.00% | **0.003** | 0.413 ± 0.015 | 3.60% | **0.010** | 0.762 ± 0.034 | 4.46% | **0.001** |
| NF | 10 | 0.444 ± 0.015 | 3.38% | **<0.001** | 0.428 ± 0.014 | 3.18% | **0.009** | 0.401 ± 0.013 | 3.29% | 0.586 | 0.721 ± 0.028 | 3.86% | 0.836 |
| NB | 9 | 0.435 ± 0.014 | 3.17% | **0.028** | 0.428 ± 0.014 | 3.25% | **0.008** | 0.412 ± 0.014 | 3.29% | **0.023** | 0.736 ± 0.028 | 3.74% | 0.186 |

**Notes.**

TH, torso height; HBL, horizontal body length; EBL, Euclidean body length; PBL, polygonal line body length; HTL, horizontal torso length.

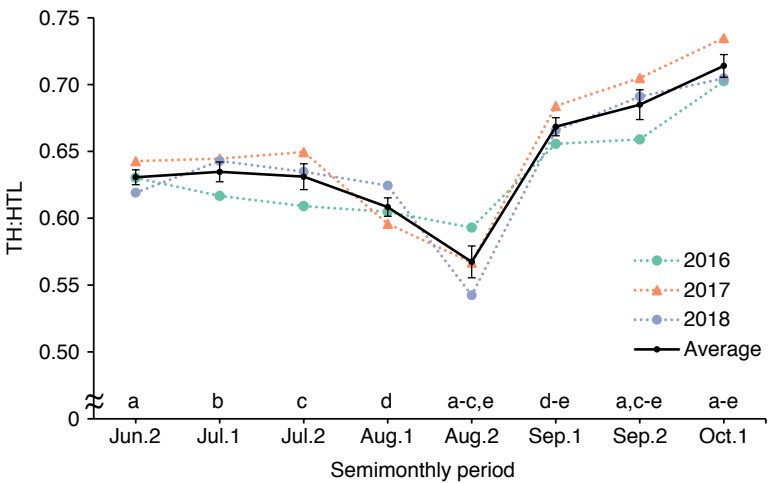

**Figure 7** **Seasonal changes in body condition estimated by calculating torso height:length ratio (TH:HTL) from photographs of an adult female brown bear in the Rusha area of the Shiretoko Peninsula, 2016–2018.** TH:HTL was compared among half-months by one-way ANOVA with a *post hoc* Tukey multiple comparison test. Same letters indicate significant differences. Error bars show SEs.

($p = 0.012$), early September ($p < 0.001$), late September ($p < 0.001$), or early October ($p < 0.001$).

## Accuracy of measurements from photographs

We captured seven adult females in the Rusha area during 2014–2016 and took photographs of each individual within 3 days before and after each capture date (Table S2). There was a positive correlation between BCI calculated from actual morphometric measurements and TH:HTL calculated from photographs ($r = 0.77$, $p = 0.042$; Fig. 8).

## DISCUSSION

We have developed a new method for visually assessing the body condition of adult female brown bears using photographs. The evaluation method consists of filtering photographs based on photograph conditions and bear posture and using photograph-based measurements of torso height and horizontal torso length in pixels to calculate the TH:HTL ratio. The significant positive relationship between TH:HTL calculated from photographs and BCI calculated from actual measurements of given individuals indicates that the body condition of brown bears can be estimated with a high degree of accuracy based on photographs. TH:HTL values increased as BCI increased, in agreement with other body condition indices, such as Quetelet's index (*Cattet, 2000*) and percent body fat (*McLellan, 2011*). This study is the first to propose a photograph-based method of evaluating bear body condition that is accurate and reliable.

The most versatile photograph-based measurement method that could be applied to bears with various postures was the measurement not of body length but of torso length. In right whales (*Eubalaena* sp.) and gray whales, body condition has been evaluated with high precision and accuracy with aerial vehicle photogrammetry by selecting photographs

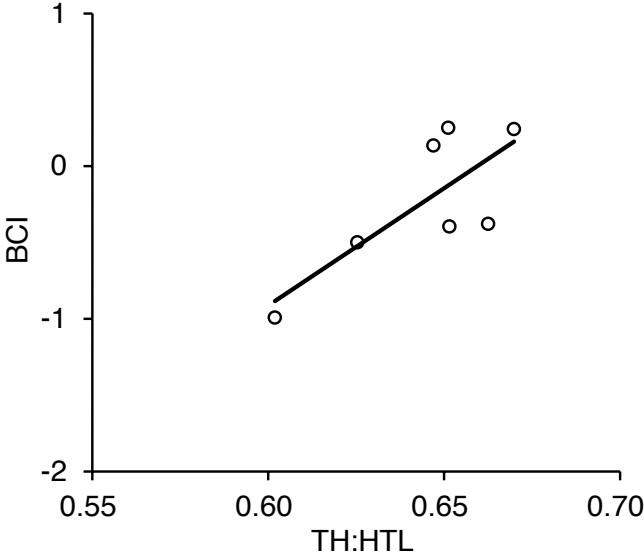

**Figure 8** Relationship between torso height:length ratio (TH:HTL) and body condition index (BCI) for seven adult female brown bears captured in the Rusha area of the Shiretoko Peninsula, Hokkaido, 2014–2016. Pearson's correlation was $r = 0.77$ ($R^2 = 0.59$, $p = 0.042$).

under strict conditions based on the whale's posture (*Perryman & Lynn, 2002*; *Christiansen et al., 2018*). However, it is not easy to collect a large number of good-quality photographs of brown bears inhabiting forests that are suitable for measurement. In fact, of the 220 photographs taken to confirm the precision of photograph-based measurement methods in this study, only 15 (6.8%) were classified into the "Good" category. Therefore, to establish a useful method of assessing body condition, it was necessary to find a method that had high applicability as well as high precision and accuracy. Although the body length of killed or captured brown bears is generally measured as the distance from the tip of the nose to the end of the last tail vertebra (*Blanchard, 1987*), in the present study all methods that included the tip of the nose in the photograph-based measurement range (i.e., HBL, EBL, and PBL) were affected by the degree of neck flexing and neck lateral bending. However, the torso length (i.e., HTL) could be measured without being affected by the condition of the neck as long as the condition of body straightness was satisfied.

TH:HTL declined from June to August and increased thereafter until the end of the field survey in early October, which suggests that bears were gaining fat over this period. The period when TH:HTL was lowest (i.e., August) coincides with the time when most cub disappearances occur in the Rusha area (*Shimozuru et al., 2017*), which indicates that poor nutrition in the summer may cause cub mortality. The seasonal changes in TH:HTL were partly consistent with BCIs calculated from killed bears, except that TH:HTL increased drastically in September. Because seasonal changes in TH:HTL were examined in only one individual in this study, it is necessary to examine how TH:HTL changes seasonally in other living bears. One factor leading to the difference between seasonal change patterns in TH:HTL and BCI may be differences in the food environment between the Rusha area and

other areas. Acorns (*Quercus crispula*), which contain large quantities of carbohydrates and fats, are a major food source throughout Hokkaido during September–November (*Ohdachi & Aoi, 1987*; *Sato, Mano & Takatsuki, 2005*). In addition, the Rusha area is considered to be a natural "ecocenter", defined by *Craighead, Sumner & Mitchell (1995)* as an area where highly nutritional food is concentrated during a certain part of the year, and many bears are present in this area to obtain these resources, in particular salmonid fish, from late August (*Yamanaka & Aoi, 1988*; *Shimozuru et al., 2017*). Therefore, bears in the Rusha area can consume higher-energy foods from late summer to autumn, which may cause their TH:HTL to increase more rapidly than the BCI of bears killed in other areas. Another possible explanation for the difference in seasonal change patterns of body condition is that most of the actual measurements were collected from bears killed for nuisance control. Throughout the lower part of the peninsula, vast agricultural farms produce mainly dent corn and sugar beets. These farms may act as an attractive sink because of the availability of human-derived foods, which lead to human-caused bear deaths (*Delibes, Gaona & Ferreras, 2001*; *Sato et al., 2011*). Therefore, there is a possibility that bears killed before September included those that had emerged into farmland or human residential areas to obtain anthropogenic foods to compensate for poor body condition. Our results suggest that including body condition data for living bears will improve estimations of seasonal and long-term trends in body condition and thus provide better estimates of the health of the bear population.

It is important to determine whether the method established using adult females in this study can be extended to other age-sex classes, other bear populations, and other bear species. Differences in body condition among age-sex classes should be taken into consideration. Our results showed that BCIs calculated from actual measurements were higher in adult males than in other age-sex classes. Therefore, relative changes in TH:HTL need to be examined by age-sex class. This study also showed no interaction between age-sex classes and seasons for BCI, which indicates that any age-sex class would show similar seasonal changes in body condition. However, it is necessary to investigate further whether the TH:HTL of other age-sex classes is able to show the seasonal changes that can be detected in adult females. Another consideration is differences in growth patterns between populations. Asymptotic body length (cm) was smaller in the Shiretoko Peninsula, 145.07 ± 1.48 and 179.47 ± 2.39 for females and males, respectively, than in two previously studied brown bear populations in northern Canada (171.55 ± 1.15 and 197.05 ± 0.69, *Bartareau, Cluff & Larter, 2011*) and Alaska (166.10–194.08 and 190.72–206.36, *Hilderbrand et al., 2018*). Therefore, when using our photograph-based method to evaluate body condition in other populations, it is necessary to select target individuals depending on the age of maturity in each population.

Because the equipment needed to weigh large-bodied animals is often inadequate or unavailable in the field, it is more difficult to directly measure the body mass of brown bears than it is to take other morphometric measurements. The TH:BL ratio measured from killed or captured bears in this study was strongly correlated with BCI, which suggests that TH:BL, as well as axillary girth, which allows us to estimate body mass (*Cattet, 1990*; *Derocher & Wiig, 2002*; *Cattet & Obbard, 2005*; *Bartareau, 2017*;

*Moriwaki et al., 2018*), can be considered a useful indicator of body condition in captured bears without direct measurement of body mass. In mice, pelvic circumference is considered a potential predictor of fat content (*Labocha, Schutz & Hayes, 2014*). In addition, abdominal girth has been widely used in measurements of humans (e.g., as part of calculating body mass index). Although torso height is a nonstandard morphometric measurement in bear studies, such additional data may make it possible to improve predictions of body condition. Furthermore, using our photograph-based method, we can overcome the technical and financial difficulties of repeated capture and can conduct periodic assessments of body condition. A noninvasive evaluation method, BCS has been previously described for polar bears (*Ursus maritimus*) (*Stirling, Thiemann & Richardson, 2008*). However, BCS is a subjective assessment system and has the disadvantage of potentially missing small changes because it uses a scale from 1 to 5. Using morphometric measurements from photographs, our method makes it possible to conduct objective and quantitative visual assessments of body condition and allows researchers to identify small fluctuations in body condition. In this study, we were able to obtain usable photographs by conducting a survey in the Rusha area, where we could photograph bears easily and safely. If automated trail cameras were installed to collect bear photographs, our noninvasive assessment method of body condition could be used widely in various locations.

## CONCLUSIONS

We developed a noninvasive method that uses photographs to assess the body condition of free-ranging brown bears and validated its accuracy against actual measurements of captured bears in the Shiretoko Peninsula, Hokkaido, Japan. Because our method is simple and applicable to photographs of bears in various postures, it can be widely applied and thus is useful for monitoring the body condition of brown bears repeatedly over the years. Using photograph-based evaluation will assist bear researchers in further investigating relationships among body condition, food habit, and reproductive success, which contribute to the conservation and management of brown bears.

## ACKNOWLEDGEMENTS

We thank H. Ose and all the fishermen engaged in salmon fishing in the Rusha area for their generous support. We are grateful to all the members of the Shiretoko Nature Foundation for their generous support. We thank all the people involved in sample collection.

### Funding

This study was conducted as the Kim-un Kamuy Project which was financially supported by AIR DO Co., Ltd. This study was financially supported by Daikin Industries, Ltd. and The Mitsui & Co. Environment Fund, Fuji Film Green Fund, and a grant-in-aid of The Inui Memorial Trust for Research on Animal Science. This research was supported by a Grant-in-Aid for Scientific Research from Japan Society for the Promotion of Science

(19K06833) and by the Environment Research and Technology Development Fund (JPMEERF20194005) of the Environmental Restoration and Conservation Agency of Japan. There was no additional external funding received for this study. The funders had no role in study design, data collection and analysis, decision to publish, or preparation of the manuscript.

## Grant Disclosures

The following grant information was disclosed by the authors:
AIR DO Co., Ltd.
Daikin Industries, Ltd.
The Mitsui & Co..
The Inui Memorial Trust for Research on Animal Science.
Japan Society for the Promotion of Science: 19K06833.
Environment Research and Technology Development Fund: JPMEERF20194005.

## Competing Interests

The authors declare there are no competing interests. Masami Yamanaka, Masanao Nakanishi, and Tsuyoshi Ishinazaka are employed by the Shiretoko Nature Foundation.

## Author Contributions

- Yuri Shirane conceived and designed the experiments, performed the experiments, analyzed the data, prepared figures and/or tables, authored or reviewed drafts of the paper, and approved the final draft.
- Fumihiko Mori and Masami Yamanaka conceived and designed the experiments, performed the experiments, authored or reviewed drafts of the paper, and approved the final draft.
- Masanao Nakanishi, Tsuyoshi Ishinazaka, Tsutomu Mano and Mina Jimbo performed the experiments, authored or reviewed drafts of the paper, and approved the final draft.
- Mariko Sashika and Toshio Tsubota analyzed the data, authored or reviewed drafts of the paper, and approved the final draft.
- Michito Shimozuru conceived and designed the experiments, performed the experiments, analyzed the data, authored or reviewed drafts of the paper, and approved the final draft.

## Animal Ethics

The following information was supplied relating to ethical approvals (i.e., approving body and any reference numbers):

All bears were captured live in accordance with the Guidelines for Animal Care and Use of Hokkaido University and all procedures were approved by the Animal Care and Use Committee of the Graduate School of Veterinary Medicine, Hokkaido University

## Field Study Permissions

The following information was supplied relating to field study approvals (i.e., approving body and any reference numbers):

Field experiments were approved by Hokkaido Regional Environment Office and Kushiro Nature Conservation Office (Permit Numbers: 1606091 and 1705182).

## Data Availability

The data are available in the Supplementary Files.

## Supplemental Information

Supplemental information for this article can be found online at http://dx.doi.org/10.7717/peerj.9982#supplemental-information.

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
