# Peer review of "Development of a noninvasive photograph-based method for the evaluation of body condition in free-ranging brown bears"

_PeerJ, doi:10.7717/peerj.9982_

## Round 0.1 · original submission · Minor Revisions

Both reviewers thought the paper needed minor revisions and I agree. Please review the comments and make the necessary edits. In addition, you should replace the use of the term "weight" with the term "mass" throughout the manuscript. As well, in Line 154, please change "bear" to "Bear". In line 216, change "BS." to "BS"., and again in line 217, change "NF," to "NF", and finally in line 218, change "NB," to "NB", and in Line 344, please change to " This study is the first to propose...".

Reviewer 1 ·

Basic reporting

No comment

Experimental design

No comment

Validity of the findings

No comment

Additional comments

I only have minor revisions to suggest- great work with this paper! It provides a novel tool that could have broad application the field. See specific comments below:

Line 52-53: Is it fat reserves that allow males to gain greater access to females, or overall body mass/ size?

Line 60 and 85: rephrase “capture surveys” to something like “capture operations”

Line 71: Brown bears are omnivores

General comment: Please define your use of “body condition” in the introduction. It is unclear if you are referring to % body fat or overall body size/mass. It becomes clearer later in the paper, but it would be beneficial to the reader to define this up front.

Lines 102-104: Consider rephrasing to: During 1998–2017, we collected body weights and
morphometrics from brown bears captured for research purposes, killed for nuisance control, or harvested from the peninsula, including the towns of Shari and Rausu (Fig. 1).

Line111-112: remove “near the tip of the peninsula” since you have a figure showing this

Line 132: rephrase to “were obtained from bears killed for nuisance control or from harvest”

Lines 154-156: Please include a justification for why you separated subadult and adult males and females into these categories. Why did you define female maturity as 5 and male as 8?

Line 377: hyphenate “higher energy”

Line 391: or bear species!

Line 402: The following could be an interesting comparison if you want to include a more recent citation for Alaska:
Hilderbrand, G. V., Gustine, D. D., Mangipane, B. A., Joly, K., Leacock, W., Mangipane, L. S., ... & Cambier, T. (2018). Body size and lean mass of brown bears across and within four diverse ecosystems. Journal of Zoology, 305(1), 53-62.

·

Basic reporting

The language was clear and professional throughout. I suggest using additional descriptive terms in addition to "measures" throughout the paper, as many times I did not understand which type of "measures" you were referring to. For example,

Your abstract needs more detail, specifically in regards to how you defined BCI. It wasn't into very far into the paper that the BCI method was explained. The statement in line 43 (short intervals over long periods of time) is confusing, and thus defining what short/long mean by example would strengthen this concept.

I appreciate the thorough review of body condition assessment methods in the Introduction. Adding the use of ultrasound measures of subcutaneous fat, as in Morfeld et al. 2014 for elephants, would strengthen this section. More general background on the wide use of body condition scoring in domestic species would be helpful (dogs, cats, etc.), as well as pigs, cattle, horses. By including these species, this strengthens the rationale of applying similar techniques to wildlife species.

Line 85 - more information why capture surveys are not suitable for ongoing body condition assessments. The term "capture surveys" is not previously introduced, which could be confusing to some readers.

The most significant challenge and confusion was the strong emphasis on correlation to BCI, but the BCI methods were not introduced until line 88, which is critical component to your paper. You should spend time explaining what this method was, citation, etc. It is very unclear as is.

Line 92 - briefly explain what "candidate methods" means.

The tables are overall very well done, with raw data submitted. However, I could not open the Supplemental files with the eps file types.

Experimental design

Overall, the design is suitable, but it did take several reads to understand the three small studies within the larger study. It would be best to state the overall goal first, and then indicate the necessary steps to get there.

Line 105: Unclear why you can determine age by teeth of some but not all of your study subjects.

The study area was nicely described. There is information on measurements in this section that should be moved, as it is not related to "study area" as the section states. (lines 117-121). Keeping the field permits in this section is good, but the animal use permit information could be moved to the animal/subject sections.

Line 141. How are the bears captured more than once, if most were killed for nuisance control/hunting as stated in line 132. It would be good to include the sample size of killed versus multiple measures in this section.

Bear capture and measurements section- include reference to the appropriate table/figure for this data.

BCI of killed or captured bears: The BCI methods should be explained more thoroughly. What did Cattet al al. find? What is the typical range of BCI? Was this developed for males, females? Please provide a review of this method.

Statistics. Having a separate statistics section would be useful, or at least define the statistics within each section. It gets confusing and difficult to read with statistical methods throughout the methods and not in a separate section or at least set apart within each section.

"Shooting and filtering of photographs". Replace the word "shooting" to something more professional (obtaining, etc.).

Lines 190-193. The supplemental file has Bear Posture, Body Arch (A) as either a score option of 1 or 3 (option 2 is blank), but the file with the photos/examples have only Score examples of 1 or 2. Is this suppose to be a Score of 3? Or why is the table "blank" for Score 2?

Results. It would be nice to see a resulting table of bears representing the different BCI categories with photos. It is still confusing to met how the BCI index works, as the table gives negative and positive values. Again, more information on this would be extremely valuable.

Discussion is very well written and addresses the significant findings of the study.

Validity of the findings

The study results are very valuable and I commend the researchers for their thorough efforts on this study. Overall, the methods and results get confusing to read, but I think separating out sections more thoroughly/accurately, as well as including a separate "Statistical Methods" section would be very helpful.

Again, explain the BCI you are utilizing throughout this paper as the gold standard for your validation. Without that level of information, it is difficult to interpret the results with clear context/meaning.

Wonderful subject, creative, and thoroughly executed, but clearer writing would help transform your robust study into more user friend action.

---

## Round 0.2 · accepted · Accept

I am happy with all your revisions.